# Spatiotemporal Patterns and Trends of Precipitation and Their Correlations with Related Meteorological Factors by Two Sets of Reanalysis Data in China

| 4  | Jinhui Jeanne Huang <sup>(1)</sup> , Nan Zhang <sup>(2)</sup> , Gyewoon Choi <sup>(2)</sup> , Edward Arthur McBean <sup>(1,3)</sup> and Qian |
|----|----------------------------------------------------------------------------------------------------------------------------------------------|
| 5  | Zhang <sup>(1,4)</sup>                                                                                                                       |
| 6  | (1) College of Environmental Science and Engineering/Sino-Canada R&D Centre on Water and                                                     |
| 7  | Environmental Safety, Nankai University, Tianjin China.                                                                                      |
| 8  | (2) Civil and Environmental Engineering, Incheon National University, Incheon, Republic of                                                   |
| 9  | Korea.                                                                                                                                       |
| 10 | (3) School of Engineering, University of Guelph, N1G 2W1 Guelph, Ontario, Canada                                                             |
| 11 | (4) Corresponding Author.                                                                                                                    |
| 12 | Corresponding author: Qian Zhang                                                                                                             |
| 13 | Sino-Canada R&D Centre on Water and Environmental Safety , College of Environmental Science and                                              |
| 14 | Engineering, Nankai University, Tianjin 300457, PR China;                                                                                    |
| 15 | Phone: (86)22-8535-8816; Fax: (86)22-8535-8816; E-mail: qianer0403@126.com                                                                   |
| 16 |                                                                                                                                              |
| 17 | Abstract                                                                                                                                     |
| 18 | This paper investigates the spatial-temporal characteristics of the changes in precipitation for China                                       |
| 19 | and the influence of other meteorological factors on precipitation. Two reanalysis datasets at monthly                                       |
| 20 | scale, namely, the GLDAS2 phase 2 forcing data 0.5×0.5 (1948 ~ 2008) and National Centers for                                                |

Environmental Prediction (NCEP) data were employed. The Mann-Kendall trend test identified the annual and seasonal changes in four meteorological factors for precipitation, air temperature, long wave radiation and surface pressure. Confidence levels of 95% were taken as thresholds to classify the significance of positive and negative trends. The trend analysis was conducted in three storm zones (I-Eastern Monsoon Region, II-Northern Inland Region and III-Qinghai-Tibetan Plateau Region) specified by Wang (2002). The findings indicate:

- Air temperature, specific humidity and downward long wave radiation, have strong correlation with precipitation, especially for the eastern monsoon region of China; while surface pressure has very weak correlation with precipitation.
- 10
  2) Latent heat shows very strong correlation with precipitation throughout China except for a
  11
  small, extremely arid area in north China where large portions of the area are deserts.
- The correlation between the volumetric soil moisture with precipitation and latent heat are
   controlled by precipitation with the characteristics of high annual precipitation and high
   correlations.
- 4) For precipitation, an increasing tendency in precipitation for the southeastern monsoon region
   and a decreasing tendency for the northeastern monsoon region (the drier region) were observed.
- Strong increasing tendencies for air temperature and downward long wave radiation, were
   observed in the northeastern monsoon region and the western area of Qinghai-Tibetan Plateau.
- Bue to changes in precipitation and air temperature and downward long wave radiation, the
   scarcity of water resources in northeastern monsoon region and flooding problems in
   southeastern monsoon region may become more severe.
- 7) The study shows that agricultural development in China may require a shift between northernand western areas to adapt to the shift in precipitation patterns.
- 24

7

8

9

Keywords: Climate Change, Mann-Kendall Trend Analysis, Correlation, Precipitation, Meteorological
 Factors

1

#### 2 1 Introduction

3 With economic development, global climate change has been become an important issue since it may have large, direct impacts on several aspects of the hydrologic cycle, in particular, bringing severe 4 5 damages as a result more frequent drought and flooding events, creating more challenging conditions 6 for managing and using water resources (Trenberth, 2010). Since precipitation is the primary source of 7 renewable water resources, changes in precipitation patterns will have substantial influence on the 8 welfare of human beings, as well as the entire ecosystems. IPCC (2014) indicates that the spatial 9 patterns of projected changes in precipitation are not uniform; for example, model projections indicate 10 that precipitation will increase by more than 1 mm/day in the southwest China, and declines will occur 11 in northern, western, and southern parts of China (IPCC WG I, Section 6.2.2).

12 Significant changes in extreme rainfall events and more frequent rainfall events have been reported 13 using the historical data over many areas in the world (e.g. Manton et al., 2001; Klein Tank and Konnen, 14 2003; Liu et al., 2005; Fujibe et al., 2005; Groisman et al., 2005; Massari et al., 2017) and in China (e.g. Gong and Wang, 2002; Gemmer, 2004; Ye et al., 2004; Li, 2011; Min, 2011; Zhu, 2011; Stephenson, 15 16 2014; Guo et al., 2017). For example, Gong and Wang (2002) have revealed significant negative 17 precipitation trends for different regions of eastern China from 1954-1998 and subsequently, positive trends from 1977 - 1998; Gemmer (2004) also observed negative precipitation trends in spring and 18 19 autumn in eastern China and positive trends in summer, and negative precipitation trends in the north 20 and north-east of China. The increasing trends of precipitation are more significant in western China, 21 particularly in the northwest (Ye et al., 2004). Most of above studies investigated the trends by the 22 ground station data. Due to limited densities of ground stations, and their abrupt variations in space, the 23 analysis and characterization of precipitation at regional scales requires reanalysis data to fill the spatial 24 and temporal gaps. This study uses two sets of reanalysis data namely, the GLDAS2 phase 2 forcing 25 data 0.5×0.5 (1948 ~ 2008) and National Centers for Environmental Prediction (NCEP) data 26 1.875×1.904 (1948 ~ 2013) to evaluate trends of precipitation levels in China.

1 Inquiry into precipitation changes under future climate has long been a research need. Statistical assessments of changes in precipitation and associated meteorological factors from the observational 2 records have provided significant evidence in understanding the changing climate (Gokmen, 2016). At 3 4 present, increased air temperatures have been widely regarded as the major factor causing climate change. Many studies have shown that temperature and precipitation have positive correlations 5 6 (Trenberth and Shea, 2005; Wu, 2012; and Zhang, 2013). However, some researchers have also 7 observed both positive and negative trends in different locations (Trenberth and Shea, 2005, Mourre et 8 al., 2016). For example, Trenberth and Shea (2005) showed that precipitation has reduced downward 9 shortwave radiation reaching the earth's surface, resulting in surface cooling which may contribute to a 10 negative correlation between precipitation and temperature.

11 In addition to temperature, some other meteorological and hydrological parameters may also have 12 substantial influences on changes in precipitation (Liu, 2009; Gong, 2006). Gong (2006) stated that 13 relative humidity was the most sensitive variable, in general, for the Yangtze River basin in China, 14 followed by shortwave radiation, air temperature and wind speed. Liu (2009) analyzed climate change 15 in Xinjiang Uygur Autonomous Region by investigating the relationship between annual precipitation 16 with mean temperature, wind speed, low cloud cover, total cloud cover, specific humidity, pan 17 evaporation, and diurnal temperature range. Lin reported that evapotranspiration is a key process in the 18 hydrologic cycle and has significant influence on precipitation. Comprehensive analyses of the 19 precipitation and its correlation with other meteorological factors are crucial to improving prediction of 20 changing climates (Fang et al., 2014).

As stated above, the present research used two sets of reanalysis data to investigate the correlation between precipitation and other meteorological and hydrological factors including downward longwave radiation, downward shortwave radiation, air temperature, specific humidity, surface pressure, wind speed, evapotranspiration, soil moisture, etc. Through correlation analysis, this study presents relationships between precipitation and many other meteorological and hydrological factors. The trends in precipitation and these meteorological and hydrological factors are also assessed in this study,

including the trend analysis in seasonal and annual scales. China has very rich varieties of topographic and climatic characteristics. The association of the correlation analysis and trend analysis with topographic and climatic characteristics are also assessed in this study. Two procedures are utilized herein to perform the correlation analysis and trend analysis as described below.

5

#### 6 2 Materials and Methods

#### 7 **2.1 Data**

8 In this study, two reanalysis datasets were used, namely, the Global Land Data Assimilation System 9 (hereafter, GLDAS) and a reanalysis dataset from National Centers for Environmental Prediction (hereafter, NCEP). For atmospheric forcing datasets, the common datasets are as follows: (1) NCEP's 10 11 Global Data Assimilation System (GLDAS); (2) NASA's Goddard EOS Data Assimilation System 12 (GEOS); (3) The European Center for Medium Range Weather Forecasting (ECMWF); (4) The 13 Princeton Global Meteorological Forcing Dataset; (5) Naval Research Laboratory Precipitation; (6) 14 NASA/GSFC TRMM 3B42RT Real-time Huffman Precipitation; (7) PERSIANN Precipitation; (8) 15 Disaggregated CMAP Precipitation; (9) Air Force Weather Agency (AFWA) Radiation; (10) 16 NOAA/CPC CMORPH Precipitation; (11) NASA/GSFC TRMM 3B42(V6) Precipitation. 17 (http://ldas.gsfc.nasa.gov/). GLDAS uses the following atmospheric metrics: 1979-1993, bias-corrected 18 ECMWF Reanalysis data (Berg et al., 2003); 1994-1999, bias-corrected NCAR Reanalysis data (Berg et 19 al., 2003); 2000, NOAA/GDAS atmospheric analysis fields; 2001-2007: a combination of 20 NOAA/GDAS atmospheric analysis fields, spatially and temporally disaggregated NOAA Climate 21 Prediction Center Merged Analysis of Precipitation (CMAP) fields, and observation-based downward 22 shortwave and longwave radiation fields derived using the method of the Air Force Weather Agency's 23 AGRicultural METeorological modeling system (AGRMET).

Global Land Data Assimilation System Version 2 (GLDAS2) dataset is bias-corrected reanalysis data, from the Terrestrial Hydrology Research Group, Princeton University. It can be downloaded at its

homepage (<u>http://hydrology.princeton.edu/</u>). The GLDAS2 data have been generated using upgraded versions of Land Surface Models (LSMs). Compared with GLDAS1, GLDAS3 has been enhanced by using the global meteorological forcing data set from Princeton University. It was produced by merging satellite and ground-based observational data products using advanced LSMs and data assimilation techniques. Its temporal coverage has been extended back to 1948.

6 GLDAS2 is a series of land surface forcing data, such as precipitation, surface meteorology and 7 radiation; state data such as soil moisture, temperature and snow; and flux data such as evaporation and 8 sensible heat flux data which were simulated by LSM. In this research, data between 1948 and 2008 9 were used with resolution of 0.5x0.5, and the meteorological factors in monthly scale are precipitation, 10 air temperature, specific humidity, downward longwave radiation, surface pressure, and wind speed.

11 The NCEP/NCAR is one kind of Physical Sciences Division (PSD) Gridded Climate Datasets. PSD maintains a collection of reanalysis datasets for use in climate diagnostics and attribution. NCEP/NCAR 12 13 Reanalysis data set (1948 - present), and it was the first of its kind of National Oceanic and 14 Atmospheric Administration (NOAA). It has been continually updated, gridded daily and monthly data 15 set that represents the state of the Earth's atmosphere, incorporating observations and numerical weather 16 prediction (NWP) model. NCEP used the same climate model that was initialized with a wide variety of 17 weather observations: ships, planes, RAOBS, station data, satellite observations and many more. It was 18 a joint product from the National Centers for Environmental Prediction (NCEP) and the National Center 19 for Atmospheric Research (NCAR). The NCEP/NCAR reanalysis mainly concentrates on using 20 initialization at a smaller scale atmospheric mode, and climate assessment. NCEP also includes Climate 21 Forecast System Reanalysis (CFSR). NCEP not only has been among the most used NCEP products in 22 history, but continued use in the future is expected with a more modern data assimilation system and 23 forecast model (Suranjana Saha, 2010). This study focused on the analysis of climate variability for a 24 set of surface variables including the monthly mean precipitation, 2m surface air temperature, surface 25 pressure, latent heat, soil moisture, upward solar radiation, downward longwave radiation, momentum 26 flux, sensible heat, and surface roughness. These data were downloaded from the National Oceanic and

Atmospheric Administration-Earth System Research Laboratory (NOAA-ESRL)
 (http://www.esrl.noaa.gov/). The temporal coverage is from 1948/01 to present, and spatial coverage is
 T62 Gaussian grid (192×94), the latitude is 88.542N ~ 88.542S, and longitude is 0E ~ 358.125E. Basic
 information for the two datasets is shown on Table 1.

#### 5 2.2 Methodology

#### 6 2.2.1 Correlation Analysis

In the theory of probability and statistics, Student's t-distribution (called t-distribution) was applied
to evaluate the mean of a symmetrically distributed population where the sample size is small and
population standard deviation is unknown.

The first dataset was from 1948 to 2008, totaling 61 years data, so the population  $n = 61yrs \times 12$ mons= 732 mons. Based on the one-sided t-distribution table, when df=730 (degree of freedom, n-2=730), for the 99% confidence level, t\* = 2.326.

13

$$t^* = r \frac{\sqrt{n-2}}{\sqrt{1-r^2}}$$
(1)

14 Following the Equation (1),  $r = \pm 0.086$ 

15

Through the Pearson product – moment correlation coefficient (Pearson's r), two variables x and y
 can be measured by the linear correlation, giving a value between +1 and -1. The formula was

$$r = \frac{\sum (x_i - \dot{x})(y_i - \dot{y})}{\sqrt{\sum (x_i - \dot{x})^2 \sum (y_i - \dot{y})^2}}$$
(2)

Based on the coefficient, r, if the value of r exceeds 0.086, it has 99% confidence level that the correlation is significant.

21

#### 1 2.2.2 Mann-Kendall Test

2 The Mann-Kendall Test (1938) has also been widely used in the meteorological field. Through use 3 of the Mann-Kendall Test, the trends of the meteorological factors, e.g. precipitation and air temperature 4 can be assessed. The Mann-Kendall test can evaluate the change tendency with long-term time series for 5 predicting the influence of potential climate change. In the past, many parametric and nonparametric 6 methods have been used for trend detection(Shi et al., 2015). Nonparametric methods usually require 7 less burdensome calculations because they are generally not related specifically to the parameters of a 8 given distribution and do not require any assumptions other than independence. The Mann-Kendall 9 trend test is a rank-based non-parametric approach that tests the randomness against trends in time 10 series datasets. It can be used to detect trends that are monotonic but not necessarily linear. The rank 11 tests are highly useful for the investigators since these tests are completed relatively quickly with an 12 efficiency of approximately 95% relative to the t-test for large size and even higher for small samples 13 (Huang et al., 2016). The Mann-Kendall trend test compares each value of time-series data with the 14 remaining values in sequential order, accounting for the number of times of increasing or decreasing. It 15 does not require the assumption of normality and only indicates the direction but not the magnitude of 16 significant trends. The Mann-Kendall trend test has a broad range of applications in hydrologic and 17 climate-related trend analysis (e.g. Tong et al., 2007, Huo et al., 2008, McBean and Motiee, 2008, Kustu 18 et al., 2010).

The null hypothesis (H<sub>0</sub>) in the Mann-Kendall test is that the data ( $x_1, x_2, x_3, ..., x_n$ ) are independent and randomly ordered. The alternative hypothesis  $H_1$  of a two-sided test is that the distributions of  $x_k$ and  $x_i$  are not identical for all k, j.

The confidence level of 95% was taken as thresholds to classify the significance of positive and negative meteorological factor trends. The trend is considered to be statistically significant if it is

- 1 significant at the 5% level (P < 0.05). The computational procedure for the Mann–Kendall test is
- 2 described as follows:
- 3 1) The entire data set consists of n data points.  $N_k$  and  $N_j$  are two sub-sets of data where the time
  - series  $x_k$  is from  $i = 1, 2, \ldots, n 1$ , and  $x_j$  from  $j = i + 1, \ldots, n$
- 5 2) Each data point  $x_i$  is used as a reference point and is compared with all the  $x_j$  data points such
- 6 that:

4

7

17

$$\operatorname{sign}(\theta) = \begin{cases} 1, \ \theta > 0\\ 0, \ \theta = 0\\ -1, \ \theta < 0 \end{cases}$$
(3)

- 8 where  $\theta = sign(k_k x_i)$ .
- 9 3) The Kendall's S-statistics is estimated by

10 
$$S = \sum_{i=1}^{n-1} \sum_{k=i+1}^{n} sign(x_k - x_j)$$
(4)

11 4) The variance for the S-statistics is determined by:

12 
$$Var(S) = \frac{1}{18} \left[ n(n-1)(2n+5) - \sum_{p=1}^{g} t_p(t_p-1)(2t_p+5) \right]$$
(5)

where g is the number of tied groups (a tied group is a set of sample data having the same value), and  $t_p$ is the number of data points in the  $p^{th}$  group. For example, for a data set of {2, 3, 5, 3, 5, 3}, it has n = 6,  $g = 2, t_1 = 2$  for the tied value 5, while  $t_2 = 3$  for the tied value 3.

16 5) The parameter  $Z_c$  is given as

$$Z_{c} = \begin{cases} \frac{S-1}{\sqrt{Var(S)}}, \ S > 0\\ 0, \qquad S = 0\\ \frac{S+1}{\sqrt{Var(S)}}, \ S 

1 where  $Z_c$  is the test statistic and follows a standard normal distribution.

The test statistic  $Z_c$  is used as a measure to identify the significance of the trend. In fact, this test statistic is used to test the null hypothesis,  $H_0$ , which means no monotonic trend in the data. If  $|Z_c|$  is greater than  $>Z_{\alpha/2}$ , where  $\alpha$  represents the chosen significance level (usually  $\alpha = 5\%$  with  $Z_{1-\alpha/2}=1.96$ ), the null hypothesis is rejected, meaning that the trend is significant with the confidence level at the magnitude of  $1-\alpha/2$  (97.5%).

The Mann-Kendall statistic S was calculated as Equation (4), the variance for the statistic S was defined by formula (5), and the test statistic Z was estimated from formula (6). A significant level is determined when |Z| > Z. In this case, the trend is considered as significant at the confidence level determined by Z (e.g. 95% when  $Z_{1-\alpha/2}=1.96$ ).

11 The Mann-Kendall test method was adopted to investigate the possible trends in monthly data series 12 for precipitation, air temperature, downward long wave radiation, specific humidity, and surface 13 pressure, sensible heat, latent heat and volumetric soil moisture.

#### 14 **2.2.3 Storm Zones**

Wang (2002) divided China into three storm zones based on the topographic features and the two precipitation extreme indicators, namely, H24m (which is the 24 hour annual maximum precipitation) and T50 (which is the annual average of days that daily rainfall was larger or equal to 50mm). The three main boundary lines are:

- 19 (1) Zone I: Southeastern side of the line along Qinling Mountains Taihang Mountains Xiao
   20 Hinggan Mountains where H24 = 70mm and T50 = 1day.
- 21 (2) Zone II: Northern margin of the line along Qinghai Tibetan Plateau where H24m < 50mm.</li>
   22 The area is called the Northern Zone below.

1

2

(3) Zone III: Western side of the line along Qinghai – Tibetan Plateau where H24m < 70mm and T50 < 1day. It is referred to as the Western Zone below.</p>

3 These zones are also referred to as, respectively, Southeast, Northern and Western China herein. The areas are shown in Figure 1 as separated by yellow lines. Zone I occupies about 45% of China and 4 has half of its area along the east coast line of China. It is also called the eastern monsoon region. This 5 6 region is strongly influenced by the monsoon climate and has the highest average annual precipitation amongst the three zones. The major storm zones are also located in this region. The north portion of 7 Zone I has lower frequencies of storms than the south portion; however, the most extreme storms in 8 9 China occur in the northern portion of Zone I (Wang 2002). Zones II and III are all inland areas. Arid 10 and semi-arid climate are the dominant climates for Zone II. Zone II is also called Northwestern arid and semi-arid region, which occupies 35% of China. Most of Zone III area is located in Qinghai-11 12 Tibetan Plateau. Qinghai- Tibetan Plateau is the highest Plateau in the world and the largest Plateau in China occupies an area of 2.3 million km<sup>2</sup>, and has an average altitude exceeding 4000 m. Due to its 13 14 extreme, high altitude and cold climate, storms rarely occur in this region and have lower precipitation than the other two regions. Zones I, II and III each have very unique topographic and climatic 15 characteristics. In this study, the correlation between precipitation and other meteorological factors as 16 17 well as the trends of meteorological factors were evaluated and analyzed based on the characteristics of 18 the three zones.

#### **19 3 Results and Discussions**

#### 20 3.1 Correlation Results

To assess the influence of various meteorological factors on precipitation, correlation analysis was conducted for China. Figure 2 shows the correlation results of precipitation with air temperature, downward long wave radiation, and surface pressure of the two datasets: GLDAS2 forcing data and NCEP data. A comparison of the results from the two datasets was also performed. Figure 3 shows the correlation results of precipitation with specific humidity of GLDAS2 and with volumetric soil