# Peer review of "Spatiotemporal Patterns and Trends of Precipitation and Their Correlations with Related Meteorological Factors by Two Sets of Reanalysis Data in China"

_Hydrology and Earth System Sciences, 2017_

## Referee Comment (RC1) · Anonymous Referee #1 · 20 Feb 2018

With the use of two reanalysis datasets (GLDAS and NCEP-NCAR), this paper first investigated the correlations between precipitation and other meteorological variables over China using Pearson correlation analysis. The paper then used Mann-Kendall test to detect the significances of trends in precipitation and six other meteorological variables at seasonal and annual scales.

General Comments:

The objective of this paper is straightforward and this paper has a clear structure with

a well-established methodology. Having said this, the basic premise of the paper is not particularly strong because the Data and Methodology sections were not well-written and there was no insightful discussions on the implication of the findings to the hydrology community (Please see the specific comments below for details). In general, there are three major questions and issues needed to be addressed in this paper:

1) Analysis based on reanalysis data

It would be a fundamental flaw to conduct correlation analysis using reanalysis data. Reanalysis data are essentially datasets derived from simulations of an atmospheric or a physical model in which the model is built with the aim to replicate the mechanisms governing the atmosphere. In other words, the model has, to some extents, already accounted for the relationships among the meteorological variables by constraining them according to some physical laws and functional relationships. Thus, examining the correlations between precipitation and other meteorological variables based on reanalysis data is not meaningful and does not add much knowledge in understanding the changes in precipitation with other variables. Also, I would be very skeptical about the results of the trend analysis because reanalysis data might contain systematic bias introduced by the model driven behind and/or might not be able to provide reliable estimates especially over the regions where observed data are very limited and highly uncertain (e.g. Tibet).

2) Justification in the use of data

The authors just randomly picked up two sets of reanalysis data for their study and they made no attempts in explaining and justifying their choice (P3:L24-26). There are increasing numbers of data available at global and regional scales (e.g. satellite-derived products, reanalysis data, and blended products from multiple sources). Why did the authors choose these two reanalysis data with different spatial resolutions for their study? Also, the result comparison of these two datasets was very shallow and there were no rigorous assessments of the performance of these two datasets over

[Figure]

China (e.g. inter-comparison of the data with in-situ observations across China). As a result, how reliable the data were over China and how confident were the authors in claiming that the data were reflecting the true relationship and trend?

3) References

There are three issues regarding the use of references in the manuscript: (a) miss-use of the references to support the statements the authors were trying to make (e.g. P3:L16: Guo et al. (2017) aimed to evaluate a new precipitation downscaling method rather than assessing the changes in extreme rainfall events in China); (b) missing references in the reference list (e.g. P4:L6: Wu (2012) and Zhang (2013) were not found in the reference list); and (c) lack of references when describing the use of data in the Data section 2.1. Please refer to the specific comments and remarks below for further information.

Specific Comments:

P2:L13-14: What did the authors mean by "high correlations" here? Please clarify.

P2:L19-23: After reading the whole manuscript, the authors did not really examine the water scarcity, flooding issues, and agricultural development over China in their study. Therefore, it is not valid to say that Points 6 and 7 are the findings of this study.

P3:L12-16: The references used here were cited wrongly. The study of Liu et al. (2005) was in China but was put as an example in the world. Stephenson (2014) was in the Caribbean but not in China. Min (2011) was a study at global scale but was put as an example in China. Also, Guo et al. (2017) aimed to evaluate a new precipitation downscaling method rather than assessing the changes in extreme rainfall events in China. This reference is inappropriate here. In addition, Gong and Wang (2002) is not found in the reference list. Please check and cite the references carefully.

P5:L8-25; P6:L1-26; P7:L1-4: The description of the data was very unclear. There were quite a lot of unnecessary information while some important information was missing.

First of all, there were no references to support the sources of the data. The web links provided in the text were just the homepage of the data providers and did not help the readers find the exact location of the data. On the other hand, the authors spent quite a decent amount of space in listing all the data sources used in GLDAS (P5:L10-17). I think this is too much of the details. Secondly, my understanding of GLDAS is there are two versions of GLDAS, namely, GLDAS-1 and GLDAS-2. GLDAS-1 combines multiple sources of datasets (as mentioned by the authors in P5:L17-23) as forcing data to drive five different land surface models whereas GLDAS-2 consists of two components: one forced entirely with the Princeton meteorological forcing data (called GLDAS-2.0), and the other forced with a combination of model and observation based forcing datasets (called GLDAS-2.1). Currently, it is very unclear in the text that which version of GLDAS the authors used in this study. I assume the authors used the latest version (i.e. GLDAS-2), but again it is unclear that whether the authors used 2.0 or 2.1. It is also very unclear to me that which land surface model outputs the authors used here as there are five different land surface models in GLDAS. Thirdly, GLDAS is provided at either 0.25° or 1.0° spatial resolutions. The authors mentioned that they used 0.5° spatial resolution. However, there were no information about how the authors re-scaled the data to 0.5°. If there was any re-scaling of this dataset, why did the authors re-scale to 0.5° but not to the same spatial resolution as NCEP-NCAR data? Fourthly, as GLDAS-2.1 only provides data from 2000 onwards, I assume the authors used GLDAS-2.0 because they mentioned that data between 1948 and 2008 were used in the study. GLADS-2.0 is driven by the Princeton meteorological forcing data. As far as I know, this Princeton dataset was constructed based on the NCEP-NCAR reanalysis, combined with a suite of global observation-based data. This means that the outputs from GLDAS-2.0 (regardless of the land surface model) would be correlated to NCEP-NCAR data to some extents and thus good agreement between GLDAS-2.0 and NCEP-NCAR would be expected. While the authors did not aim to inter-compare or evaluate the performance of these two dataset over China, it would be a bit redundant to present the same analysis with two similar products. Please

consider re-writing the whole section by providing succinct background information of the data and stating clearly which version of the data the authors used in the study.

P7:L7-20: The logical flow of this section is a bit odd to me. The main subject of this section should be Pearson correlation coefficient because the authors wanted to conduct a correlation analysis. However, the authors first presented the Student's t-distribution (without any references) and came up with the t-statistics, and put the Pearson correlation coefficient at the end without explaining clearly the linkage between them. How does Student's t-distribution relate to Pearson correlation coefficient when conducting a correlation analysis? I suggest the authors re-writing this section by first introducing Pearson correlation coefficient [Equation (2)]. Then they could proceed to mention Pearson's r has a t distribution with n-2 degree of freedom and the test statistic is given by Equation (1). After that, they could explain how to get a statistical significant value of r at the 99% confidence level.

P8:L2-18: This section requires a bit of polishing. First, there was no references provided for the use of Mann-Kendall test. Secondly, the statement in L4-5 was invalid because the Mann-Kendall test is not used for predicting the influence of potential climate change. Third, what did the authors mean by "less burdensome calculations" in L7? Lastly, the reference used in L13 (Huang et al. 2016) did not really support the statement made in L10-12 because Huang et al. (2016) did not use Mann-Kendall test in their study and did not make any comments about the efficiency of using Mann-Kendall test and t-test. Please clarify.

P9:L8-10: The notations of the equations are confusing. Shouldn't be $\theta=(x\_k-x\_j)$ instead of $\theta=sign(k\_k-x\_j)$? Also, shouldn't be "j" instead of "i" in the Equation (4)? Please check the equation and correct me if I were wrong.

P11:L20: The whole section 3.1 could be better written. Instead of just spelling out the results variable by variable, product by product, and region by region, the authors should also provide discussion on possible reasons, for instance, why such spatial

patterns occurred in this region, or why such relationship occurred over this part of the region, or why the relationship shown by this product was different, if any, from the other one. Currently, the whole section was merely a summary report on what could be seen from the figures without any substantial discussion on the results.

P12:L4-21: I am a bit skeptical about the results presented here because precipitation, air temperature, and downward longwave radiation by themselves have seasonality. I suspect that the positive correlations here were greatly affected by the seasonal cycles of the time series, which are quite obvious and well-known from a climatology perspective. In addition, I think the authors should provide a table of summary statistics on the ranges of correlation in each storm zone instead of merely saying some very qualitative statements such as "stronger correlation", "very similar fashion", "very good agreement", etc. when comparing the results of GLDAS-2 and NCEP-NCAR.

P13:L9-27; P14:L1-6: This is very surprising to me because all of a sudden the authors mentioned that the latent heat flux was one of the focus of the study (L15-16). If latent heat flux was one of the focuses in this study, the authors should explicitly state this objective in the Introduction section but not here! The authors should also provide a literature review and a discussion on why it is important to look at the correlation between latent heat flux and other meteorological variables. What are the knowledge gaps in examining latent heat flux and other meteorological variables? Accordingly, the title of this manuscript should also be changed to reflect this.

P15:L15-24: Comparison of the results from the two datasets here could be problematic because of the differences in the spatial resolutions. NCEP-NCAR has a coarser spatial resolution than GLDAS-2. This means that the spatial heterogeneity is averaged out over a larger area and results in a smoother surface of the precipitation field. Conducting a trend analysis over two different spatial resolutions and comparing the spatial pattern directly might not be appropriate as such spatial pattern could be scale dependent. Moreover, the authors should provide explanations for the regions where opposite tendencies were witnessed from the two datasets. Why did the two datasets

result in different trends? What could water resource managers do given such results?

P16:L1: The whole section 3.2.2 suffers similar problems aforementioned: 1) results were reported variable by variable, zone by zone, and product by product; 2) very shallow comparison was done to discuss the similarities and differences between the two datasets; and 3) assessments were done based on qualitative descriptions. What are the main messages of this section? Can the increasing or decreasing trends be quantified in a summary statistics for each storm zone?

P19:L8: The discussion and implication of this section 3.3 is very general and vague. The authors concluded the changes in agricultural practices in each storm zone by merely linking the results from the trend analysis to a land use map. Such conclusions were over-simplified the complicated mechanisms and relationships among precipitation, hydrological changes, irrigation practices, and water resources management. Besides, this section was quite incoherent with the rest of the manuscript because the authors did not mention at all they were going to discuss the results with agricultural practices in China in the Introduction section.

P20:L20-21: This conclusion was over-stated. This study did not really examine anything related to agriculture development in China.

P21:L13: What did the authors mean by "different temporal scales"? Please clarify.

P23:L1: Please update the reference list, especially for those references published in HESS. They have already become final papers in HESS but currently they were still referenced as HESS-D paper.

Remarks:

P1:L20: missing degree sign, it should be "0.5°x0.5°" not "0.5x0.5"

P3:L6: wrong reference, it should be "Trenberth, 2011" not "Trenberth, 2010"

P4:L6: missing reference of Wu (2012) and Zhang (2013) in the reference list

P4:L12: missing reference of Gong (2006) in the reference list

P4:L17: inappropriate in-text citation: "Lin reported that . . ."

P4:L24-25: This statement is a repetition of the statement in P4:L21-24. Please consider deleting it.

P5:L11: should be "GDAS" not "GLDAS"

P6:L2: should be "GLDAS2" not "GLDAS3"

P7:L4: Table 1 is not found.

P8:17-18: missing reference of Kustu et al. (2010) in the reference list

P11:L4: should be "red" lines not "yellow" lines

P12:L25: very high "latitudes"?

P13:L11: missing reference of Liang et al. (2014) in the reference list

P14:L14: should be "refers to December" not "refer December"

P18:L18: should be "Figure 9" not "Figure 6"

Figure 1: the authors should provide the elevation in the legend and provide information about the storm zoning here. The figure should be stand-alone without the need to refer to the text.

Figures 2 to 9: These figures require significant improvement. First of all, the authors should mask out the data outside China because this study focused on China only. There are no points in showing the data results in India, Southeast Asia, and over the oceans! Also, the authors should crop out the regions between 0° and 15°N and make each sub-figure best fit to the extent of China only. Otherwise, showing these extra regions and extra data results will distract the attention of the readers from interpreting the results. Actually, it makes a very hard time for me to read the figures (especially Figures 5 to 9 in which I could hardly identify where China is). Secondly, the space

of the figures could be better utilized. Use one colour bar to indicate the correlation coefficient as one legend and put it either at the bottom or on the right-hand side of the figures. There is no needs to show the colour bar for each sub-figure because the colour bar is essentially showing the same information! Thirdly, I suggest the authors revise the colour scheme they used in Figures 2 to 4 by changing the white colour into other colour because white is usually used to represent no data or missing data. Also, the sub-title in each sub-figure in Figure 2 is redundant because the information has already been shown on the left-hand side of each row.

---

## Referee Comment (RC2) · Anonymous Referee #2 · 24 Feb 2018

This paper describes a study on the distributions of the correlations of climate variables (precipitation, temperature, ...) and trends of individual variables over China based on two reanalysis data sets. A wide range of climate variables have been analysed over the whole country. The topic is within the remit of HESS, and should be of interest to the HESS community. However, the paper has several weak areas that should be addressed:

Major issues 1) Since there are many global data sets available, please justify why GLDAS2 phase 2 and NCEP data sets are selected in this study (e.g., any evidence

that they are more suited to China than other data sets); 2) There are a variety of ways to divide China into different zones. The zones should try to be homogeneous for the study purpose. It would be useful if the paper could describe what zoning options are available in China, and why the 3-Zone approach (with H24m and T50) by Wang (2002) is selected in the study. It would be also useful to explain if the decision was correct in view of the results. IT is possible that a better zooning system could be developed; 3) For the hydrological community, precipitation and evapotranspiration (ET) are the main concerns instead of other variables such as air temperature, long wave, radiation, surface pressure, etc. It would be useful to add potential ET and actual ET in the study; 4) Since reanalysis data sets are generated by computer models, it would be useful to validate the results using the ground observations (e.g., at some selected points); 5) It seems that the authors are concerns about the agricultural development and its link with the shift in precipitation patterns. It would be useful to show the history of land use/land cover in China instead of just one snap shot (e.g., LULC maps over several decades) so that the agriculture development trend can be compared with the precipitation trend. The trends of Potential PE and actual PE could also be considered in assessing the agricultural impact by climate change.

Minor issues: 1) It is useful to mask out the area outside of China 2) One legend is sufficient for each figure. 3) Although the paper is generally well written, some minor typos/grammatical errors should be checked and removed.

---

## Editor Comment (EC1) · J. Hanesiak (Editor) · 28 Feb 2018

The editor has received two thoughtful reviews of your article titled "Spatiotemporal Patterns and Trends of Precipitation and Their Correlations with Related Meteorological Factors by Two Sets of Reanalysis Data in China". Both reviewers suggest acceptance but with major revisions.

Some of the major comments include:

Reviewer #1 (R1):

The basic premise of the paper is not particularly strong because the Data and Methodology sections were not well-written and there was no insightful discussions on the implication of the findings to the hydrology community. Three specific major comments are:

1) Justification of correlating reanalysis variables that were derived from the same reanalysis system (see R1 specific comment)

2) Justification of using the two reanalysis datasets that were chosen for the study (both reviewers commented on this). Please explain why these two datasets were chosen over others and if they are better than others over China. Please see the specific comment from R1.

3) Three issues with respect to references throughout the manuscript. See specific comment from R1.

Reviewer #2 (R2)

1) Explanation of why the three major China sub-division zones were selected and whether there is another potential approach for this.

2) Inclusion of potential evapotranspiration (ET) and actual ET in the correlation analysis, if possible.

3) Simple surface point validation of reanalysis datasets for random selected points.

4) Illustrate the history of land use/land cover in China instead of just one snap shot so that the agriculture development trend can be compared with the precipitation trend, and potential ET/actual ET if possible.

Both reviewers also had very good suggested "Specific Comments" and "Minor Issues" that should be considered in the revised manuscript.

If the authors are willing to revise the manuscript and explain how each of the above points were addressed by both reviewers, the article will be considered again for publication. Both reviewers are willing to review the article again once the edits have been made.
* * *

---

## Editor Comment (EC2) · J. Hanesiak (Editor) · 10 Apr 2018

Thank you to the author's for sending in the revised manuscript. Both reviewers (R1 and R2) agreed to review the revised manuscript one final time to ensure the article has addressed all comments. I would ask both reviewers to examine the replies from the authors and review the revised manuscript to ensure all edits have been made. Once I have received the reviewers 2nd comments, the manuscript will be considered for final acceptance, if no further edits are necessary. I thank both reviewers for their thoughtful insights.

---

## Author Comment (AC1) · 10 Apr 2018

Firstly, the authors express our appreciation to your review of the paper. Your careful review has resulted in a greatly improvement of our paper. Our reply and comments are as follows:

1. Reply to your question about analysis based on reanalysis data

1) We used the reanalysis data from the websitebelow http://hydrology.princeton.edu/data.lsm.php. It has been renamed as "Global Land

[Figure]

Surface Model Data", and now requires registration to obtain the data set.

2) Reanalysis data is obtained based on observation data and has been used widely as a supplementary data to observation data, especially in ungagged area. A very important example is the widely used Statistic Downscaling Tool, SDSM, which use NCEP reanalysis data as the ground truth to do regression with a large set of Metrologic parameters.

3ïijĽThe investigation in this study did not present a stable strong correlation with all the metrological parameters; a large variation has been shown in the study.

2. Reply to your question about justification in the use of data

The two-setsreanalysis data were not randomly picked up. First of all, NCEP reanalysis data as said before, has been widely accepted (e.g. being used in SDSM as the ground truth observation) and has been well studied; it is however, with coarse resolution. While GLADS dataset might be less investigated in the past, it however is with high resolution of 0.5°ïĆ'0.5°. The two dataset are both from US and may share a similar observational data source and the structure of calibration. It is then might be easier to compare the discrepancy induced by different resolution. Error exists everywhere. By comparing the results from two dataset, it is a way to reduce the error. In addition, the GLDAS has also been used previously to study the climate change in the world, e.g. Gomez (2018), Jon Gottschalck (2005) et al. and in China, e.g.Y Y Liu (2012); we then choose the two datasets for further investigation in our study.

3. Mistakes about references

The detailed mistakes about references have been fully corrected in the manuscript using redline edit. Also, some minor spelling and grammatical errors have been corrected, and we have masked out the area outside of China in each figure.All the corrections are in the supplement below.
Please also note the supplement to this comment:
https://www.hydrol-earth-syst-sci-discuss.net/hess-2017-756/hess-2017-756-AC1-supplement.zip

―――――――――――――――――――――――

---

## Author Comment (AC2) · 10 Apr 2018

Firstly, the authors express our appreciation to your review of the paper. Your careful review has resulted in a greatly improvement of our paper. Our reply and comments are as follows:

1. Reply to your question about why GLDAS2 phase 2 and NCEP data sets are selected in this study.

The two-setsreanalysis data were not randomly picked up. First of all, NCEP reanalysis

data as said before, has been widely accepted (e.g. being used in SDSM as the ground truth observation) and has been well studied; it is however, with coarse resolution. While GLADS dataset might be less investigated in the past, it however is with high resolution of 0.5°ïĆť0.5°. The two dataset are both from US and may share a similar observational data source and the structure of calibration. It is then might be easier to compare the discrepancy induced by different resolution. Error exists everywhere. By comparing the results from two dataset, it is a way to reduce the error. In addition, the GLDAS has also been used previously to study the climate change in the world, e.g. Gomez (2018), Jon Gottschalck (2005) et al. and in China, e.g.Y Y Liu (2012); we then choose the two datasets for further investigation in our study.

2. Reply to your question about why the three major China sub-division zones were selected.

The partition of sub-division zones in China was based on a 50-year study since 1950's funded by Ministry of China Water Resources. It was published in a book entitled"Torrential Rainfall in China" by Jiaqi Wang in 2002. It now became the national guideline for design and planning.

3. Reply to your suggestion about add potential ET and actual ET in the study.

The calculation of PET for grid datainvolves a lot computation. It can be another study. It is then very hard to be included in the current study. So does actual ET.

4. Reply to your suggestion about add simple surface point validation of reanalysis datasets for random selected points.

The validation by randomly selected ground meteorological station data was included in the revised manuscripts. The source of data was added in P7L12-16 and Figure 1, the correlation analysis was added in P15L1-7 and Table 2, the Mann-Kendall analysis was added in P17L7-10, and P21L1-6 as well Table 3. The validation based on the ground station indicates the similar trend and correlation with the results from the two

reanalysis data sets.

5. Reply to your suggestion about illustrate the history of land use/land cover in China.

We have tried a lot of efforts to search for the land use maps. The land use map over entire China is not free of charge; the cost of obtaining these maps is substantial. Additionally, evaluating the history of land use change in china and comparing with the precipitation trend and PET/Actual ET could be a good research topic and a good paper itself, while it would involve a lot of efforts. Thanks for the idea and we may apply some additional funding for this research in the near future, it however could not be done in this paper. It seems to be too much to be integrated into one paper.

6. Reply to your minor question.

The detailed mistakes about some minor spelling and grammatical errors have been corrected, and we have masked out the area outside of China in each figure and corrected the legend. The detailed correction are in the supplement below.

Please also note the supplement to this comment:
https://www.hydrol-earth-syst-sci-discuss.net/hess-2017-756/hess-2017-756-AC2-supplement.zip

---

## Author Comment (AC3) · 10 Apr 2018

Dear editor:

The authors express their appreciation to the reviewers of the paper. Their careful review has resulted in a greatly improved paper. According to the editorial opinion, the introductory part has been extensively revised, and many other details have been revised one by one. And all the new corrections are in the suplement below, check it please, thank you.

[Figure]

Please also note the supplement to this comment:
https://www.hydrol-earth-syst-sci-discuss.net/hess-2017-756/hess-2017-756-AC3-
supplement.zip
* * *

---

## Editor Comment (EC3) · J. Hanesiak (Editor) · 4 May 2018

Dear Dr. Wong. Thank you very much for your thoughtful and helpful review of the manuscript titled "Spatiotemporal Patterns and Trends of Precipitation and Its Correlations with Related Meteorological Factors by Two Sets of Reanalysis Data in China" by Huang et al. The authors have submitted their revised manuscript and I kindly ask that you please review this latest version, since you agreed to do so before the article is formally published. Please ensure all edits are to your satisfaction, or provide one final set of suggested edits. Thank you for your time and effort with this.

[Figure]

Best Regards John Hanesiak